SOFTWARE

# PseudoknotVisualizer: Visualization of pseudoknots on three-dimensional RNA structures

Takumi Otagaki *, Goro Terai, Kiyoshi Asai, Junichi Iwakiri

Department of Computational Biology and Medical Sciences (CBMS), The University of Tokyo, Kashiwa, Chiba, Japan

* takumiotagaki@gmail.com

## Abstract

**Summary:** We introduce the PseudoknotVisualizer, a specialized software designed to identify and visualize pseudoknots within RNA three-dimensional structures. Typically, RNA secondary structures containing pseudoknots can be decomposed into multiple pseudoknot-free layers. Our software colors the base pairs in each pseudoknot layer, enabling the visualization of pseudoknot distribution within three-dimensional structures. Specifically, users can utilize the PseudoknotVisualizer as a PyMOL extension, applying it directly to RNA molecules loaded in PyMOL. Additionally, a Command Line Interface (CLI) is provided, allowing users to generate coloring commands in Chimera or PyMOL formats, which can then be manually copied and pasted for visualization. By facilitating the clear depiction of pseudoknots in RNA tertiary structures, this tool addresses significant challenges in the identification and visualization of pseudoknots in RNA structural analysis, thereby enhancing research productivity and expanding potential applications in molecular biology.

**Availability and implementation:** PseudoknotVisualizer is freely available at https://github.com/TakumiOtagaki/PseudoknotVisualizer.

## Author summary

In this study, we introduce PseudoknotVisualizer, an open-source tool for direct visualization of RNA pseudoknots within three-dimensional structures. RNA pseudoknots are critical for functions such as telomerase activity and programmed ribosomal frameshifting, yet are difficult to identify and display in 3D viewers. PseudoknotVisualizer offers two user-friendly interfaces—an extension for the world's most widely used 3D molecular graphics viewer and a command-line interface compatible with that viewer or another leading platform. Both interfaces leverage external modules that extracts base-pairing information from tertiary

**Data availability statement:** Source codes and manuals have been deposited at the GitHub repository: https://github.com/TakumiOtagaki/PseudoknotVisualizer.

**Funding:** This work was supported by Japan Society for the Promotion of Science (JSPS) KAKENHI Grant Numbers JP24H00737, JP22H04925(PAGS) to K.A., and JP23K28183, JP25H01166 to J.I.; and Japan Science and Technology Agency (JST) CREST Grant Number JPMJCR23N1 to K.A.. The funders had no role in study design, data collection and analysis, decision to publish, or preparation of the manuscript.

**Competing interests:** The authors have declared that no competing interests exist.

coordinates of RNA structures and then iteratively decomposes these base pairs into pseudoknot-free layers via dynamic programming. Each layer is assigned a unique color, enabling rapid, intuitive recognition of pseudoknot distribution in complex RNA molecules. Applying our tool to 1,915 RNA chains from the PDB revealed that canonical base pairs predominate in the core layer (87 %), while non-canonical base pairs are enriched (30 %) in the other non-nested layers; most chains exhibit no more than three layers of pseudoknots. By clarifying pseudoknot architecture in RNA 3D structures, PseudoknotVisualizer accelerates structural analysis and supports the development of computational structure prediction methods.

## 1 Introduction

Pseudoknots are structural motifs formed by base pairing between two single-stranded regions separated by an intervening stem; their base pairs are non-nested interactions (they cross when drawn in two dimensions) [1,2]. In many contexts, the term "pseudoknot" is the subset of the non-nested interactions involving only canonical base pairs (Watson–Crick [3] and wobble [4]). The rest of the non-nested interactions involve non-canonical base pairs [5–7]. These non-nested interactions, including pseudoknots, are integral to various RNA tertiary formations and play biologically significant roles, such as promoting telomerase activity within the RNA component of telomerase [8,9] and serving as programmed ribosomal frameshift motifs in several RNA viruses [10–12]. Due to the labor-intensive process of determining RNA tertiary structures, the number of RNA structures recorded in databases is only about 3.80 % compared to that of proteins (https://www.rcsb.org/stats/explore/polymer_entity_type, access date: 2025/08/31). This gap underscores the need for highly accurate RNA tertiary structure prediction methods; yet, accurate prediction of pseudoknotted structures remains a considerable challenge. In the CASP15 [13] structural prediction competition, approximately 70 % of RNA targets contained pseudoknots, and high-accuracy predictions were rarely achieved, suggesting that RNA tertiary structure prediction involving pseudoknots is intrinsically challenging.

Several applications exist for analyzing pseudoknots: some can generate 2D diagrams of secondary structures with pseudoknots [14–16], while others identify pseudoknot formation based on the base-pairing information [17,18]. Among these software, RNApdbee (and its updated version RNApdbee 2.0) [15,16] gives users a choice of base-pair annotators and multiple algorithms for the pseudoknot layer decomposition and presents their calls in a common format; users can obtain dot-bracket strings including pseudoknots for downstream analysis.

Nevertheless, seamless integration of pseudoknot-aware annotation with interactive 3D visualization within a single workflow remains limited. This can hinder rapid inspection of spatial context and long-range interactions when studying pseudoknots, which is important for RNA tertiary structure analysis, including prediction.

To address this need, we developed the PseudoknotVisualizer, a tool that visualizes pseudoknots within RNA tertiary structures. This software allows users to input RNA tertiary structure data in the PDB or mmCIF format. It identifies pseudoknots (and other non-nested interactions involving non-canonical base pairs) and clarifies their spatial layouts in the widely used 3D molecular viewers, Chimera [19] and PyMOL [20]. We expect this tool to enhance researchers' intuitive understanding of pseudoknots within RNA structures and to accelerate developments in RNA tertiary structure prediction methods by providing clear and immediate visualization. At the same time, accurate RNA secondary-structure prediction—particularly in the presence of pseudoknots—remains difficult; recent benchmark analyses across Machine Learning (ML) and non-ML methods report variable, often modest accuracy [21].

## 2 Design and implementation

As shown in Fig 1, PseudoknotVisualizer offers two primary interfaces: a PyMOL extension and a Command Line Interface (CLI). The PyMOL extension works within PyMOL, while the CLI supports Chimera and PyMOL.

Both interfaces rely on external modules; RNAView [17] and DSSR [18,22,23] to extract base-pairing information from PDB or mmCIF files. Once RNAView or DSSR is installed, users can analyze an RNA object and a specified chain in PyMOL by entering `pkv RNAobject[, chainID, annotator]`. Since RNAView and DSSR adopt different geometric criteria and differ in how base-pairing interactions are handled, their outputs can legitimately diverge for the same input. At the same time, multiple strategies exist for pseudoknot-layer decomposition, and different choices can lead to different decompositions (i.e., different layerings).

Each layer is colored differently to enhance visual distinction, making it easier for researchers to quickly identify and interpret the locations of pseudoknots. For large RNA molecules, where intuitively identifying pseudoknots can be particularly challenging, PseudoknotVisualizer simplifies this task by providing clear and distinct color visualization and enables immediate identification of pseudoknots, as demonstrated in Fig 2. In addition to coloring, the PyMOL extension creates separate PyMOL selections for the different layers, which enables users to interactively analyze pseudoknots.

The CLI provides a method to use PseudoknotVisualizer without the need for a graphical user interface. By executing the following command, users can generate a coloring script for a specific chain within a PDB or mmCIF file.

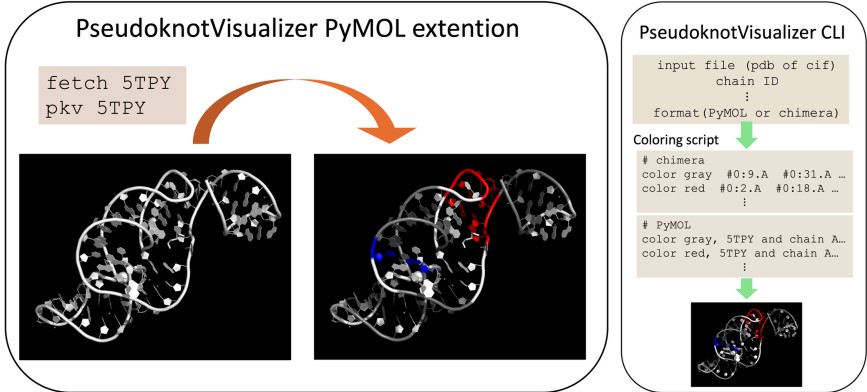

**Fig 1**. **Workflow of the PseudoknotVisualizer.** The software offers two usage options: a PyMOL extension and a Command Line Interface (CLI). The left figure illustrates the PyMOL extension, where a tertiary structure loaded into PyMOL (in this case, 5TPY) is analyzed and visualized by entering `pkv 5TPY` in the command line. This command applies color to the pseudoknot layers, with red indicating pseudoknot layer 1 and blue indicating layer 2, while additional layers (if present) can be visualized using other colors. Note that the entire molecule was pre-colored white to highlight the red and blue pseudoknot layers. The right figure demonstrates the CLI workflow. Like the PyMOL extension, the CLI takes a PDB or mmCIF file as input and generates a script for coloring in Chimera or PyMOL. By pasting this script into the command line of the specified viewer, users can visualize pseudoknot layers by applying colors to the structure.

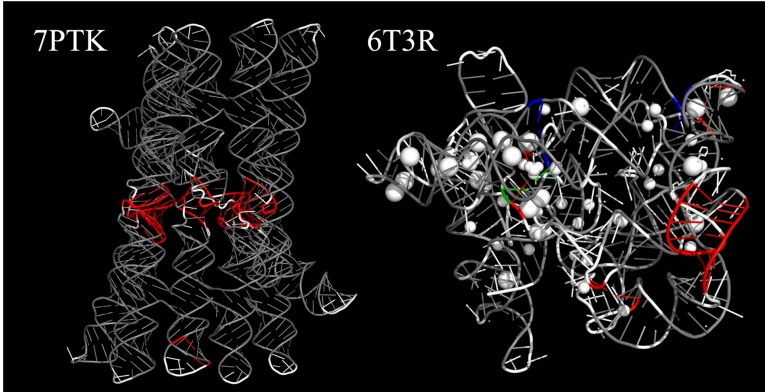

**Fig 2**. **Examples of the structures which have non-nested interactions involving canonical base pairs and non-canonical base pairs visualized by PseudoknotVisualizer are shown.** This figure illustrates examples from PDB IDs 7PTK [27] and 6T3R [28], both of which are large sequences exceeding 300 nucleotides in length, making them particularly challenging to analyze visually without computational assistance. Note that, as demonstrated in Fig 1, all structural elements were set to white prior to applying the PseudoknotVisualizer coloring operation for clarity. These visualizations were generated with `annotator=RNAView` and the option `include_all`, thereby including the non-nested interactions involving non-canonical base pairs. The color scheme is as follows: white represents unpaired bases; gray indicates paired bases including non-canonical base pairs in the core layer; red, blue, and green represent paired bases in non-nested layers 1, 2, and 3, respectively.

```
python PseudoknotVisualizer/CLI_PseudoknotVisualizer.py \
-i /path/to/your_pdb.pdb \ # Supports PDB and mmCIF formats.
-o /path/to/output_script.txt \
-a RNAView \ # or DSSR (annotator)
-c chain_id \
-f pymol
```

The generated script can be configured for Chimera compatibility by using `-f chimera` option and then pasted into the PyMOL or Chimera command line to apply the coloring. For detailed usage and implementation, the source code for PseudoknotVisualizer, along with installation instructions and example datasets, is publicly available on GitHub at https://github.com/TakumiOtagaki/PseudoknotVisualizer.

A brief explanation of how PseudoknotVisualizer operates is as follows: it begins by extracting base pair information from the user-provided 3D structural data using the external module RNAView or DSSR. Based on the complete list of base pairs, it identifies sets of pseudoknot-free base pairs, which are either mutually nested or exhibit bifurcation relationships between their components. Using dynamic programming, PseudoknotVisualizer determines the subset with the maximum number of base pairs. Once this optimal set is identified, it is removed from the initial list of base pairs. This process is iteratively repeated until the original list of base pairs is empty [24]. Through this iterative approach, the algorithm systematically extracts sets of pseudoknot-free base pairs, ordering them by the number of base pairs they contain. Consequently, the distinct sets of base pairs identified in this manner are inherently related as pseudoknots. These sets are then sequentially assigned to layers, starting with layer0 (core layer) for the largest set, followed by layer1, and so on (pseudoknot layer), ensuring a clear hierarchical representation of the base pair sets. The number of these layers is called pseudoknot order [25]. Further algorithmic details are provided in S2 Text. As related work, similar layered dynamic programming strategies have been used in RNA structure prediction, notably CaCoFold, which first derives a maximally nested structure under covariation-based positive/negative constraints and then incorporates remaining (potentially non-nested) interactions in subsequent layers [26]. Our layering instead operates on base pairs annotated from 3D coordinates and optimizes the size of a non-crossing subset for visualization. We chose this greedy algorithm, which maximizes

the number of base pairs at each step. In practice, this naturally maximizes the number of base pairs in the core layer, thereby minimizing those assigned to the pseudoknot layers. This, in turn, helps users to visually understand the spatial distribution of pseudoknots. In addition, the algorithm is not only intuitive but also supported by its successful application in CaCoFold [26]. While we selected this strategy for these advantages, we note that alternative algorithms could yield different results. We can also extend this algorithm to all non-nested interactions, including pseudoknots; in that case, we refer to the resulting layers as non-nested layers rather than pseudoknot layers because pseudoknot is used only for the non-nested interactions involving only canonical base pairs (see Introduction).

## 3 Results

Using the pseudoknot layer decomposition algorithm described in Sect 2, we decomposed 1,915 chains from RNA-solo [29] into pseudoknot-free layers (filters: repository=BGSU, redundancy=all class members, molecule=solo RNA, method=all, resolution $\leq$ 4.0Å; accessed 2025-07-25). RNAsolo provides pure RNA structures, deriving from Protein Data Bank. Canonical base pairs were defined as Watson–Crick pairs (A–U, G–C) and wobble pairs (G–U) [5]. While our default visualizations used canonical pairs, we also report analyses that include non-canonical pairs. Among these 1,915 chains, after excluding entries with no base pairs/layers, the analyzable set sizes were: RNAView (canonical only) 1,726 chains, of which 703 had at least one pseudoknot layer; RNAView (all pairs) 1,789 chains, of which 756 had at least one non-nested layer; DSSR (canonical only) 1,718 chains, of which 687 had at least one pseudoknot layer; DSSR (all pairs) 1,727 chains, of which 788 had at least one non-nested layer. Using DSSR, we obtained 44,698 canonical and 8,331 non-canonical base pairs; using RNAView, 48,306 canonical and 6,509 non-canonical pairs. These differences reflect distinct geometric criteria and treatment of nucleotide interactions; however, the key trends reported below were consistent across both annotators.

Fig 3a shows the distribution of the number of non-nested layers (when considering only canonical base pairs, this corresponds to the pseudoknot order), annotating with DSSR. With canonical pairs only, most chains have $\leq$ 3 layers; including non-canonical pairs slightly shifts the tail, but orders >4 remain rare. The observed distribution of pseudoknot orders is consistent with findings from previous studies [15], and RNAView results show the same overall trend despite some differences (Supplementary S1 Fig).

We then extracted chains which have at least one non-nested layer. Fig 3b summarizes, for each layer, the per-chain fraction of non-canonical versus canonical pairs. The layer0 (core layer) is dominated by canonical pairs, whereas non-nested layers tend to show a higher fraction of non-canonical pairs. RNAView analysis again yields a broadly similar pattern, with some deviations from the DSSR results (Supplementary S1 Fig). While these results are robust across the two annotation tools, it should be noted that adopting a different decomposition algorithm could potentially lead to variations in the observed distributions.

## 4 Availability and future directions

Our present work provides an overview of base pairs which form the non-nested interactions, including pseudoknots. Our next step is to conduct a comprehensive statistical analysis of these interactions to better understand RNA structure. Specifically, we will quantify the distribution of the types of base-pairs which form the non-nested interactions and structural contexts and assess their contributions to 3D structure folding. Insights from these analyses will guide updates to PseudoknotVisualizer, which is expected to support structural prediction and analysis.

As shown in the Results section, the different combination of base-pair annotators and pseudoknot decomposition algorithms might lead to different results. In order to give various options to users, we are planning to add the following options and update; (i) *Base-pair annotators.* We plan to support other annotators (e.g., MC-Annotate, FR3D, ClaRNA

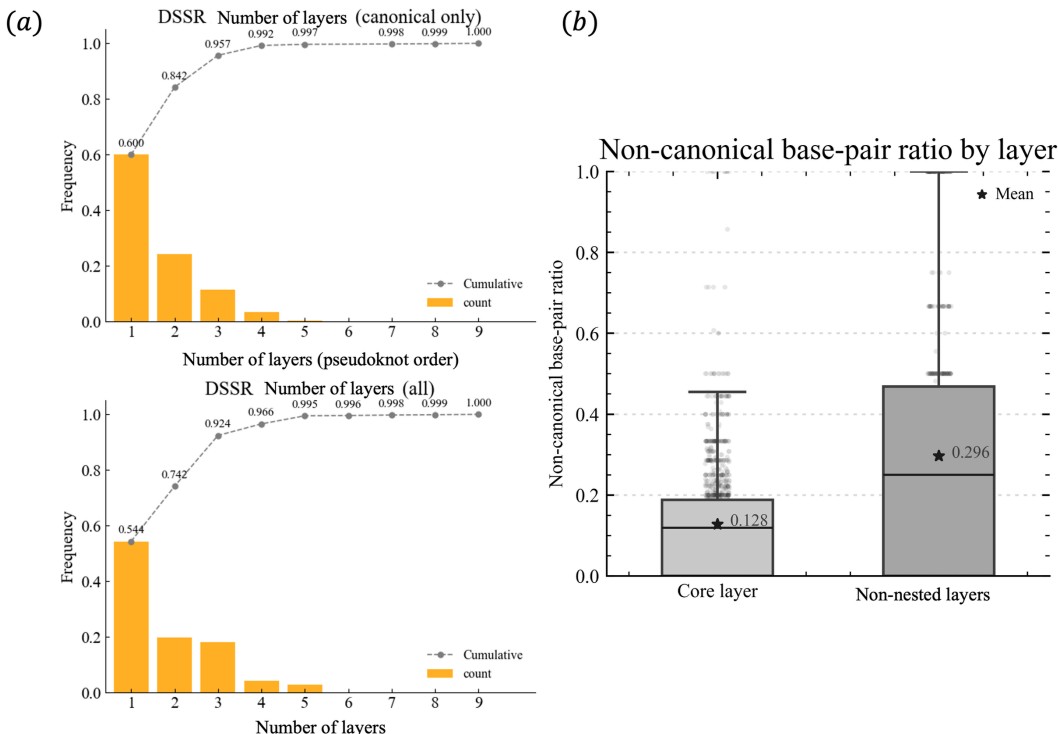

*(a)*

*(b)*

**Fig 3**. **(a) Both graphs represent the frequency of the number of layers in RNA chains from PDB entries on the condition that these entries contain at least one base pair.** The top graph considers only canonical base pairs, showing that over 80 % of chains have at most 2 layers. The bottom graph includes all base pairs, including non-canonical ones, demonstrating a larger number of layers compared to the top graph. Nevertheless, over 80 % of chains still have at most 3 layers. (b) Distribution of canonical and non-canonical base pairs across different decomposed layers in RNA molecules that contain at least one pseudoknot or other non-nested interactions, as determined by the PseudoknotVisualizer. Results shown were obtained using DSSR as the base-pair annotator.

[7,30,31]), as user-selectable back ends. This will make base-pair calls more comprehensive and allow side-by-side comparisons across annotators. (ii) *pseudoknot decomposition algorithms.* In addition to the current maximum-cardinality non-crossing subset, we will provide alternative algorithm proposed by [24], so that users can select the criterion that best matches their analysis.

**Availability:** PseudoknotVisualizer is freely available at https://github.com/TakumiOtagaki/PseudoknotVisualizer. Installation instructions and usage examples are provided.

## 5 Conclusion

The PseudoknotVisualizer software facilitates the visualization and observation of pseudoknots in the tertiary structures of RNA. This advancement is expected to address challenges related to the prediction of pseudoknotted structures. Accurate prediction and visualization of pseudoknots are essential for effective analysis. Qualitative comparisons of predicted structures, which are also important for accurate prediction, are difficult to perform automatically, often requiring manual inspection of structural features. In such cases, visualizing the pseudoknots in a 3D viewer becomes especially important for interpreting RNA structural features accurately. Furthermore, this tool accelerates efficiency for researchers in conducting detailed RNA 3D structure analyses, enabling faster interpretation of data and discovery of structural insights.

## Supporting information

**S1 Text. Base-pair detection methods (base-pair annotators).** DSSR `v1.9.10-2020apr23`; RNAView `v2.0.0 (Jan 2024)`.
(PDF)

**S2 Text. Details of pseudoknot layer decomposition algorithm.** We render the drawing in layers. Given the allowed base pairs $E$, we perform greedy layering by a Nussinov-style dynamic programming pass that extracts a maximum-cardinality non-crossing subset at each layer. Each DP pass runs in $O(n^3)$ time and $O(n^2)$ space. This strategy is effective for visualization but does not guarantee the minimum number of layers.
(PDF)

**S3 Text. Data and code availability.** Code: https://github.com/TakumiOtagaki/PseudoknotVisualizer (license `MIT`). The scripts to recreate the graphs are provided. The full chain list (PDB ID + chain) is available at `analysis/dataset/pdbid_chains.txt` in the repository.
(PDF)

**S1 Fig. Supplementary analyses.** (a) Distribution of the number of layers when using RNAView as the base-pair annotator. The trend is consistent with the DSSR-based results; including non-canonical base pairs increases the number of layers compared with using canonical pairs only. (b) Distribution of canonical and non-canonical base pairs across layers. Non-nested layers have a higher fraction of non-canonical base pairs than the core layer.
(PDF)

## Acknowledgments

We thank the attendees of the RNA Informatics Dojo 2024 in Hokkaido, Japan for valuable discussions.

## Author contributions

**Conceptualization:** Takumi Otagaki, Goro Terai, Kiyoshi Asai, Junichi Iwakiri.

**Data curation:** Takumi Otagaki.

**Formal analysis:** Takumi Otagaki.

**Funding acquisition:** Kiyoshi Asai, Junichi Iwakiri.

**Investigation:** Takumi Otagaki, Junichi Iwakiri.

**Methodology:** Takumi Otagaki, Goro Terai, Junichi Iwakiri.

**Project administration:** Takumi Otagaki, Junichi Iwakiri.

**Resources:** Takumi Otagaki, Kiyoshi Asai.

**Software:** Takumi Otagaki.

**Supervision:** Goro Terai, Kiyoshi Asai, Junichi Iwakiri.

**Validation:** Takumi Otagaki, Junichi Iwakiri.

**Visualization:** Takumi Otagaki.

**Writing – original draft:** Takumi Otagaki.

**Writing – review & editing:** Takumi Otagaki, Goro Terai, Kiyoshi Asai, Junichi Iwakiri.

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
