## [Decision Letter · Decision Letter 0]

7 Jul 2025

PCOMPBIOL-D-25-01111

PseudoknotVisualizer: Enabling Direct Visualization of Pseudoknots on Three-Dimensional RNA Structures

PLOS Computational Biology

Dear Dr. Otagaki,

Thank you for submitting your manuscript to PLOS Computational Biology. After careful consideration, we feel that it has merit but does not fully meet PLOS Computational Biology's publication criteria as it currently stands. Therefore, we invite you to submit a revised version of the manuscript that addresses the points raised during the review process.

Please submit your revised manuscript within 60 days Sep 06 2025 11:59PM. If you will need more time than this to complete your revisions, please reply to this message or contact the journal office at ploscompbiol@plos.org. Please include the following items when submitting your revised manuscript:

We look forward to receiving your revised manuscript.

Kind regards,

Shi-Jie Chen

Academic Editor

PLOS Computational Biology

Arne Elofsson

Section Editor

PLOS Computational Biology

**Journal Requirements:**

2) Your manuscript is missing the following sections: Design and Implementation, Results, and Availability and Future Directions. Please ensure that your article adheres to the standard Software article layout and order of Abstract, Introduction, Design and Implementation, Results, and Availability and Future Directions. For details on what each section should contain, see our Software article guidelines:

https://journals.plos.org/ploscompbiol/s/submission-guidelines#loc-software-submissions

Potential Copyright Issues:

i) Figure 1. Please confirm whether you drew the images / clip-art within the figure panels by hand. If you did not draw the images, please provide (a) a link to the source of the images or icons and their license / terms of use; or (b) written permission from the copyright holder to publish the images or icons under our CC BY 4.0 license. Alternatively, you may replace the images with open source alternatives. See these open source resources you may use to replace images / clip-art:

2) If any authors received a salary from any of your funders, please state which authors and which funders..

**Reviewers' comments:**

Reviewer's Responses to Questions

**Comments to the Authors:**

**Please note that one of the reviews is uploaded as an attachment.**

Reviewer #1: The paper by Otagaki et al. presents PseudoknotVisualizer, a visualization plugin designed for PyMOL and Chimera that highlights pseudoknot-forming base pairs in RNA 3D structures using distinct colors. The tool integrates existing third-party algorithms to detect base pairs and identify pseudoknots of various orders, and then applies color coding to these elements within a rendered 3D model. While the plugin is of some practical use for visualizing RNA pseudoknots, the work appears to be primarily an engineering-level implementation effort, suitable for an undergraduate-level project. The core computational tasks - base pair annotation and pseudoknot classification - are delegated entirely to external tools, with the authors contributing the integration of these tools and visualization layer. Scientifically, the novel aspect of the paper lies in the descriptive statistics of canonical and noncanonical base pairs across different structural layers and pseudoknot orders. Notably, the plugin does not support the currently recommended mmCIF input format or new-style PDB identifiers, which significantly limits its applicability to the newest and large RNA structures. These are my main concerns regarding the manuscript. Additional comments are outlined below.

(1) The Introduction opens with the sentence: "Pseudoknot structures are complex RNA formations where two or more groups of base pairs form between distinct loop regions of the other base pairing groups." First, the phrase "regions of the other base pairing groups" is unclear and should be rephrased for precision. More importantly, this definition of pseudoknots appears to be overly narrow. Pseudoknots form when base-pairing interactions occur between two single-stranded regions separated by a stem, and these single-stranded regions do not necessarily have to be loops - they can include other unpaired segments such as extended single-stranded tails. If the authors define pseudoknots strictly as interactions between loops, does this imply that their software cannot handle cases where pseudoknots involve a loop and a non-loop single-stranded region? Clarification is needed both in the definition and in the scope of the tool's capabilities.

(2) In the second paragraph of the Introduction, the authors state: "However, there are no useful tools that can efficiently identify and clarify pseudoknots in 3D RNA structure data." This statement is inaccurate and somewhat surprising, especially considering that the authors themselves cite RNApdbee, a well-established tool that directly addresses this task. RNApdbee analyzes RNA 3D structures (!), annotates base pairs including pseudoknots, and generates 2D structural visualizations in which pseudoknots of different orders are highlighted with distinct colors and symbols in an extended dot-bracket notation. The tool allows users to choose among multiple algorithms (four for base-pair annotation and five for topological classification of pseudoknots) and supports modern data formats, including mmCIF. Although RNApdbee does not provide 3D visualization like PseudoknotVisualizer, it offers significantly broader functionality for the analysis, annotation, and topological decomposition of pseudoknots in RNA 3D structures. I recommend that the authors revise this claim to reflect the actual state of available tools and clarify how their approach complements or differs from existing solutions.

(3) The authors have arbitrarily selected specific tools for base-pair annotation (Yang et al., 2003) and pseudoknot topology decomposition (Smit et al., 2008), but they do not justify why these particular methods were chosen. Are the authors aware that multiple algorithms exist for both base-pair detection and pseudoknot classification, and that these tools can yield substantially different results? In fact, the same 3D RNA structure can be interpreted as having very different maximum pseudoknot orders depending on the methods used. For example, in the case of PDB entry 3J7Q, the identified pseudoknot order may vary between 4, 5, 6, or even 7 when analyzed with different algorithms available in RNApdbee.

The current implementation of PseudoknotVisualizer relies on a fixed pipeline, using tools preselected by the authors, without informing the user that these choices affect the outcome or that alternative tools might lead to different interpretations. This lack of transparency is problematic, as it may mislead users into assuming that the presented results are deterministic and definitive. I strongly recommend that the authors either provide a rationale for their choices or, preferably, implement options for algorithm selection or include a warning regarding result variability.

(4) At the end of the Introduction, the authors write that their script might enhance researchers’ intuitive understanding of pseudoknots within RNA structures and to accelerate developments in RNA tertiary structure prediction methods by providing clear and immediate visualization. The authors may also consider acknowledging the current limitations in RNA secondary structure prediction when pseudoknots are involved. While several algorithms exist that attempt to predict secondary structures including pseudoknots, the accuracy of such predictions remains low. This limitation is well illustrated in recent analyses, such as the study by Justyna et al. (2023), which highlights the challenges associated with reliable prediction of RNA secondary structures with and without pseudoknots.

(5) "Consequently, the distinct sets of base pairs identified in this manner are inherently related as pseudoknots. These sets are then sequentially assigned to layers, starting with layer0 for the largest …" - regarding this fragment, it would be helpful to clarify explicitly that layer0 refers to the core structure, i.e., the base-pairing layer that does not contain any pseudoknots. While this may be implied, stating it directly would improve clarity for readers who are less familiar with pseudoknot decomposition convention used in this work.

(6) The sentence "Using the pseudoknot layer decomposition algorithm described in Section 2" appears to refer to a description of the algorithm that is actually presented in the unnumbered section titled Main Features, which either functions as part of the Introduction or was unintentionally left without a section number. In the current structure of the manuscript, Section 2 contains statistical analyses rather than a description of the decomposition algorithm. I recommend correcting this reference to avoid confusion and to ensure accurate navigation for the reader.

(7) "After excluding complexes with proteins or other molecules for simplicity and selecting only structures composed solely of RNA, we obtained a dataset of 1,788 RNA chains" - Since the content of the PDB is constantly updated, I recommend that the authors specify the exact date of data retrieval to ensure reproducibility. As an alternative source for curated RNA-only 3D structures, the authors may also consider referring to the RNAsolo database (https://rnasolo.cs.put.poznan.pl/), which facilitates the extraction of RNA chains not involved in complexes with proteins or ligands.

(8) In the sentence "These 1,355 chains contained a total of 24,047 canonical base pairs and 7,198 non-canonical base pairs" it is unclear whether these statistics refer to the entire structures or only to the regions involved in pseudoknot formation. I recommend that the authors clarify this point in the manuscript.

(9) In the caption of Figure 3, the phrase "number of layer" should be corrected to "number of layers".

(10) In Figure 3(b), the two adjacent plots use different vertical axis scales: the left plot ranges from 0 to 0.5, while the right plot ranges from 0 to 0.35. Using inconsistent y-axis scales in side-by-side plots is a classic form of statistical misrepresentation and makes it difficult to accurately compare the data between the two charts. I strongly recommend adjusting the figure so that both plots use the same y-axis scale to enable proper visual comparison.

Reviewer #2: This manuscript introduces a visualization tool to directly identify non-nested RNA base pairs within 3D structural renderings of all atoms of an RNA structure, such as PyMOL.

My main point about this manuscript is about the use of the word pseudoknot. I believe, they should be called non-nested interactions. Pseudoknots are a particular subset of non-tested interactions that involve Watson-Crick base pairing only. The rest of the non-nested interactions involve non-WC base pairing.

The distintion is important, because pseudoknots consist of nested WC base pairs the stack forming helices are subject to the same evolutionary presures than the nested WC helix and also display covariation when homologous pseudoknots are aligned. The same is mostly not true for the rest of non-nested non-WC base pair interactions, which do not form helices.

-- For instance this definition in the paper

"Pseudoknot structures are complex RNA formations where two or more groups of base pairs form between distinct loop regions of the other base pairing groups."

should be replaced byt "two or more groups of WC base pairs". And overal the non-nested interactions includes pseudoknots and all other non-WC base pairs.

-- Also the abstract statement

"canonical base pairs predominate in the main layer (78.6 %), while pseudoknot layers are enriched in non-canonical pairs (50.4 %);"

should be re-written as "non nested layers include about ~50% pseudoknots versus other non-canonical pairs".

it would be interesting to see what kind of nested base pair they are seeing in the main layer that are not WC that constitute more than 20% of them.

--- The title should also be re-written.

-- Also, the dp algorithm to select layers is not described. Also notice that using a layered DP to select maximal subsets of nested interactions has been used extensively before (example, CaCoFold).

Reviewer #3: See attached PLOS-PseudoknotVisualizer.pdf

**Have the authors made all data and (if applicable) computational code underlying the findings in their manuscript fully available?**

Reviewer #1: Yes

Reviewer #2: Yes

Reviewer #3: Yes

PLOS authors have the option to publish the peer review history of their article (what does this mean?). If published, this will include your full peer review and any attached files.

Reviewer #1: No

Reviewer #2: No

Reviewer #3: **Yes: **Xiang-Jun Lu

**Figure resubmission:**
---

## [Decision Letter · Decision Letter 1]

5 Oct 2025

PCOMPBIOL-D-25-01111R1

PseudoknotVisualizer: Visualization of Pseudoknots on Three-Dimensional RNA Structures

PLOS Computational Biology

Dear Dr. Otagaki,

Thank you for submitting your manuscript to PLOS Computational Biology. After careful consideration, we feel that it has merit but does not fully meet PLOS Computational Biology's publication criteria as it currently stands. Therefore, we invite you to submit a revised version of the manuscript that addresses the points raised during the review process.

Please submit your revised manuscript within 30 days Dec 05 2025 11:59PM. If you will need more time than this to complete your revisions, please reply to this message or contact the journal office at ploscompbiol@plos.org. Please include the following items when submitting your revised manuscript:

We look forward to receiving your revised manuscript.

Kind regards,

Arne Elofsson

Section Editor

PLOS Computational Biology

Arne Elofsson

Section Editor

PLOS Computational Biology

**Journal Requirements:**

**Reviewers' comments:**

Reviewer's Responses to Questions

**Comments to the Authors:**

Reviewer #1: I appreciate that the authors positively responded to all my comments. I am aware theat it required quite a lot of effort but the text is much more clear now. I do not have any other questions/comments.

Reviewer #2: This new version has made some clarifications, but they still insist on a redefinition of the term pseudoknot that is absolutely unnecessary and conflicts with the actual definition of pseudoknot (the arrangement of two helices of WC base pairs where one of the helices involves residues in the loop of the other helix, [Batey, Rmabo, Doubna, 1999]), and ignores all the literature on other RNA 3D architectures in the RNA loops involving nonWC base pairs (Westhof, Leontis).

Reviewer #3: The authors have addressed my comments and concerns.

**Have the authors made all data and (if applicable) computational code underlying the findings in their manuscript fully available?**

Reviewer #1: Yes

Reviewer #2: Yes

Reviewer #3: Yes

PLOS authors have the option to publish the peer review history of their article (what does this mean?). If published, this will include your full peer review and any attached files.

Reviewer #1: No

Reviewer #2: No

Reviewer #3: No

**Figure resubmission:**
---

## [Editor Report · Decision Letter 2]

2 Nov 2025

Dear Mr. Otagaki,

We are pleased to inform you that your manuscript 'PseudoknotVisualizer: Visualization of Pseudoknots on Three-Dimensional RNA Structures' has been provisionally accepted for publication in PLOS Computational Biology.

Best regards,

Shi-Jie Chen

Academic Editor

PLOS Computational Biology

Arne Elofsson

Section Editor

PLOS Computational Biology

---

## [Editor Report · Acceptance letter]

PCOMPBIOL-D-25-01111R2

PseudoknotVisualizer: Visualization of Pseudoknots on Three-Dimensional RNA Structures

Dear Dr Otagaki,

I am pleased to inform you that your manuscript has been formally accepted for publication in PLOS Computational Biology. Your manuscript is now with our production department and you will be notified of the publication date in due course.

With kind regards,

Anita Estes
